# Towards an Understanding of the Effect of Adding a Foam Core on the Blast Performance of Glass Fibre Reinforced Epoxy Laminate Panels

**DOI:** 10.3390/ma14237118

**Published:** 2021-11-23

**Authors:** Sherlyn Gabriel, Christopher J. von Klemperer, Steeve Chung Kim Yuen, Genevieve S. Langdon

**Affiliations:** 1Blast Impact and Survivability Research Unit (BISRU), Department of Mechanical Engineering, Univer-sity of Cape Town, Rondebosch 7701, South Africa; steeve.chungkimyuen@uct.ac.za; 2Department of Mechanical Engineering, University of Cape Town, Rondebosch 7700, South Africa; chris.vonklemperer@uct.ac.za (C.J.v.K.); genevieve.langdon@sheffield.ac.uk (G.S.L.); 3Department of Civil and Structural Engineering, University of Sheffield, Sheffield S1 3JD, UK

**Keywords:** foam core sandwich panels, fibre reinforced polymers, blast loading, failure, transient response

## Abstract

This paper presents insights into the blast response of sandwich panels with lightweight foam cores and asymmetric (different thicknesses) glass fibre epoxy face sheets. Viscously damped elastic vibrations were observed in the laminates (no core), while the transient response of the sandwich panels was more complex, especially after the peak displacement was observed. The post-peak residual oscillations in the sandwich panels were larger and did not decay as significantly with time when compared to the equivalent mass laminate panel test. Delamination was the predominant mode of failure on the thinner facesheet side of the sandwich panel, whereas cracking and matrix failure were more prominent on the thicker side (which was exposed to the blast). The type of constituent materials used and testing conditions, including the clamping method, influenced the resulting failure modes observed. A probable sequence of damage in the sandwich panels was proposed, based on the transient displacement measurements, a post-test failure analysis, and consideration of the stress wave propagation through the multilayered, multimaterial structure. This work demonstrates the need for detailed understanding of the transient behaviour of multilayered structures with significant elastic energy capacity and a wide range of possible damage mechanisms. The work should prove valuable to structural engineers and designers considering the deployment of foam-core sandwich panels or fibre reinforced polymer laminates in applications when air-blast loading may pose a credible threat.

## 1. Introduction

Explosions cause devastating consequences such as loss of life and massive structural damage. They remain high in the public consciousness due to high profile accidents [1] and terrorist attacks [2,3] in the recent past. Some of these events have targeted transportation system [4], making blast loading an unfortunate threat to structures comprising fibre-reinforced polymers (FRP) laminates or sandwich panels.

Lightweight materials are increasing in popularity as they offer mass savings (and therefore the potential of improved fuel economy or increased speed) for transportation applications, for example, car bodies, bicycle frames, and marine and aerospace structures. Fibre reinforced polymers are particularly attractive as they have tuneable structural properties (through careful design of the fibre orientation and lay-up, resin system and manufacturing process). Glass FRPs (GFRPs) are among the most popular materials for structures, as they possess good thermal and corrosion properties in addition to their high-specific strength and stiffness, often at a lower cost than their carbon fibre-based rivals.

Sandwich structures offer the potential to increase the structural stiffness of FRP laminates by introducing a lightweight core that increases the second moment of area of the sandwich. Such structures comprise thin but relatively stiff outer face sheets and a lightweight core. However, the overall strength of the sandwich is often limited by the strength of core, and the thin face sheets (while advantageous weight-wise) may result in a weak sandwich panel. Tekalur et al. [5] stated that maximising the strength of the core, skin, and interface between the two was a better alternative compared to simply increasing the thickness (and weight) of the core and skin.

Previous research investigated the dynamic response of sandwich panels with polymeric foam cores where the face sheets were balanced (i.e., of equal thickness). Arora et al. [6] investigated the response of full-scale sandwich panels to far field air blast loading. The sandwich panels were constructed using GFRP skins and a styrene foam core. Front face cracking (skin fibre breakage) and localised delamination around the cracked region were evident. Shear cracking in the core and interfacial failure between the front face and core were also observed. The back face, however, remained intact with no cracking or tearing. Wang and Shukla [7] also found no damage on the back face sheet on E-glass Vinyl Ester face sheet/styrene foam core sandwich panels subjected to a uniform blast load via a shock tube facility. Buckling and failure in the front face sheet, facilitated by in-plane compressive loading, was observed and believed to reduce the blast resistance of the sandwich composites.

Gupta and Shukla [8] investigated the effect of temperature on the blast performance on marine foam core sandwich composites and found that the dynamic response and failure mechanisms corresponded to the high strain rate behaviour of the individual constituents at a given temperature. Failure mode maps for lightweight composite panels subjected to blast loading were generated by Andrews and Moussa [9] with the assumptions of large stand-off distances, simply supported boundary conditions and symmetric face sheets. The failure modes considered were core failure (due to shearing), face sheet wrinkling and face sheet failure due to bending stresses. While it can provide some insight on these possible failure modes for various designs, it does not include core crushing and delamination.

For blast tests conducted at small stand-off distances (resulting in more localised loading response), failure modes such as core compression, core cracking and fragmentation, debonding between the core and face sheets, delamination, and fibre rupture and complete penetration were observed by Langdon et al. [10,11]. It was found that front face sheet rupture would occur before plastic deformation and debonding of the back face sheet, and complete core penetration [11]. Furthermore, sandwich panels with denser cores had lower levels of damage. Additionally, equivalent laminates were compared to the sandwich panels and found to have not reached penetration failures at the same impulses [10].

The damage is influenced by several factors, particularly the stand-off distance of the explosive charge, material properties and configuration of the individual constituents. Additionally, the front face sheet influences the blast resistance of the sandwich panels. It was stated that the damage to the front face sheet is more detrimental to the performance of a sandwich panel than the damage to the back face sheet [12]. A comprehensive review of sandwich panels and laminates can be found in Wanchoo et al. [13], which details the influence of the core and face sheets, and effect of stand-off distance.

The failure of the sandwich panels appears to follow soon after the front face sheet damage. Thus, thicker face sheets are often used to improve the overall performance of the panel, substantially increasing the panel mass, especially if thicker face sheets are used on both sides of the sandwich structure. In this work, to limit the weight implications of thicker face sheets, the blast response of sandwich panels with asymmetric face sheets (with a thicker face sheet on the blast side) was studied experimentally. While there are studies on the blast response of asymmetric metal sandwich panels [14,15,16], little is known on the asymmetric sandwich panels with polymer foams and FRP face sheets. This paper reports on the experimental results of these asymmetrical sandwich panels with E-glass fibre reinforced epoxy face sheets and closed cell PVC foam cores subjected to air-blast loading. A uniformly distributed load was imparted onto the panels. The transient response and failure progression with increasing impulse was investigated from the experimental results. The blast response was compared to that of FRP laminates using the same glass fibre system.

## 2. Materials

### 2.1. Laminate and Face Sheet Materials

Glass FRPs have high tensile strength and chemical resistance. When combined with the epoxy resin, as used in previous blast studies [17,18], the composites exhibited good resistance to localised air blast loading. As a control for the plain laminate systems, 19 layers of fabric were used and laid up [0/90] to produce panels with an approximate mass of 1 kg (300 mm × 300 mm panel area) after infusion. A 400 g/m^2^ plain weave glass fibre system, pre-treated with silane [19], was used as it was readily available and commonly used within South Africa.

Prime 27 LV (AMT Composites, Cape Town, South Africa), and its predecessor Prime 20 LV, are low-viscosity, petroleum-based epoxy resin systems that are suitable for use in resin infusion manufacturing processes [20,21]. It is used for yacht hulls, masts, and wind turbine blades. When used with a slow hardener it is suitable for the manufacture of large parts or thick sections, having low exothermic characteristics.

### 2.2. Sandwich Construction

The number of plies for the face sheets was designed to be 2:1 for the front to back face sheet ratio. The sandwich panels had 18 total layers of the same glass fibre system, where 12 layers was used for the front face and 6 layers for the back face of the composite as shown in Figure 1. This was chosen to give an approximate equivalent mass to the 19-layer laminates.

### 2.3. Sandwich Core Materials

The sandwich foam cores were closed cell PVC foam with a nominal density of 80 kg/m^3^ (tradename Divinycell H80, DIAB Group, Gloucester, UK), cut from 25 mm thick large sheets with an approximate uniform density and isotropic microstructure. The same core material was used in sandwich structures by Langdon et al. [10,11] subjected localised loading conditions. However, this work differed in that the face sheets were asymmetric and uniform blast loading conditions were applied.

The reported compressive strength for H80 was 1.4 MPa, correlating with the manufacturers’ claimed technical data shown in Table 1 [22] and those of Ye et al. [23]. The plateau stress is known to increase with increasing strain rate [23].

## 3. Manufacturing

All panels for the blast testing and material characterisation experiments were manufactured from the constituent materials using the Vacuum Infusion (VI) technique. The sandwich panels were made in a single step VI manufacturing process similar to Ref. [11]. The required numbers of layers of woven fabric were laid up on either side of the dry foam core (with release plies and flow promotion media enclosing the overall dry sandwich panel configuration) and enclosed in a vacuum bag while a vacuum was drawn to enable resin flow and ensure compaction. The laminates were manufactured the same way without the core as shown in Figure 2.

The top surface temperature of the panels was monitored to ensure it stayed below 50 °C. Once the infusion was completed and the part had cured in the mould, the flow promotion materials and release films were removed. The manufactured panels were cut into smaller panels and post-cured at 50 °C for 16 h according to the resin manufacturer’s recommendations [20,21]. The ramp rate for increasing and decreasing temperature was 1 °C per minute. Once postcuring was completed, the panels were inspected for visual defects such as large cracks, or dry spots of fabric. Defective parts were discarded. The blast test and material characterisation specimens were then cut from the panels, at least 50 mm away from the free edges of the large panels and 20 mm away from any other edges in the smaller ones (to ensure thickness consistency). The dimensions of the panels were measured after manufacture. The 19-layer GFRP laminate panels had an average thickness of 6.2 mm and a mass of 1.02 kg. The sandwich panels had an average thickness of 30.8 mm and average mass of 1.21 kg. The mass of the sandwich panels was unfortunately higher than the laminates, however, reducing the number of plies while maintaining the designed face sheet ratio would have theoretically resulted in a lower mass than 1 kg. Given that previous work showed that the sandwich panel performance was dependant on front face sheet thickness, it seemed prudent to aim at a slightly higher mass than a slightly lower mass. In hindsight, different number of laminate layers would were used, but this proved impractical within the current study.

## 4. Material Properties

### 4.1. Flexural Properties

Quasi-static 3-point flexural tests were performed on rectangular strips of the laminates at a constant cross-head speed of 3 mm/min, width: thickness ratio of 13:4 and a span: thickness ratio of 16:1 in accordance with ASTM 7264 [24]. At least five repeat tests were performed. The primary concern was to confirm the consistency of the manufactured specimens, and to provide properties for future modelling and simulation work. The resulting stress-strain curves are shown in Figure 3. The response of the laminates was initially linear followed by a sharp drop in strength after the peak flexural stress was reached, indicating the material had very limited plastic capacity (as expected). The laminates failed on the compression side, with surface buckling and delamination under the central loading point.

The same span:thickness ratio of 16:1 was used to test rectangular strips of the asymmetrical sandwich panel with the average width of 30.5 mm. Tests were performed using the same cross head speed as the one used for the laminates. The thicker skin side was placed directly under the centre loading nose as the thicker face sheet was chosen to face the blast.

A schematic of the stress distribution of the sandwich beam under pure bending is shown in Figure 4 with the assumption that a linear variation of the strain through the thickness of the neutral axis. The core contribution was ignored as the modulus of the core E_c_ was likely much smaller than face sheets E_f_ and E_b_. The differences in modulus would have caused the neutral axis to be closer to the top surface, and consequently, the stress on the top surface would be lower than on the bottom surface. Despite this, damage was only observed on the top surface by the loading nose, and no damage was observed on the bottom surface. This suggests that the compressive limit for the top face sheet was lower than the tensile limits of the bottom face sheet with the reduced thickness.

In contrast to the laminates, the response of the sandwich specimens was initially quasi-linear, up to 25 MPa, then the strength appeared to plateau and dropped as shown in Figure 5. The force-displacement curve taken over a longer time span is shown in Figure 6. According to Daniel and Abot [25], the bending behaviour of a sandwich panel is governed by the face sheets. Thus, it can be expected that the failure of the sandwich panels would occur due to compressive failure of the top skin, which was evident based on the failure observed on the beams.

A summary of the results for flexural properties of the laminates and smeared properties of the sandwich specimens are presented in Table 2.

### 4.2. Tensile Properties of the Laminates

Quasistatic, in-plane, tensile tests were performed on the GFRP laminates. The tests followed the procedure in ASTM D 3039 [26], with specimens of 25 mm nominal width, 250 mm nominal length, and a 2.5 mm nominal thickness. Fewer layers (12 instead of 19) were used in the specimen manufacture to achieve the desired 2.5 mm thickness. Tensile tests were performed on specimens with their longitudinal axis at 0°, 45°, and 90° relative to the woven fabric warp direction. The crosshead speed was kept constant at 1 mm/min and at least five tests per orientation were performed. To facilitate mounting in the grips and prevent slippage, the ends of the specimens were wrapped with emery cloth. The face of the tensile specimen was painted with a speckle pattern for filming. Digital image correlation was used to record the surface strain during the test.

A results summary is presented in Table 3. As expected, there were minimal differences between the 0/90° orientations, but the 45° orientation specimens exhibited greater ductility, lower elastic moduli and lower ultimate tensile strength. The dominating failure modes in the 45° GFRP specimens were fibre shifting and pull-out (as well as shear) rather than a pure tensile response. The standard deviations were low, indicating consistency in the manufacturing process.

## 5. Blast Testing

### 5.1. Experimental Arrangement

The 300 mm × 300 mm panels were clamped along all four sides, leaving a square exposed area of 200 mm × 200 mm. The front clamp frame was integral with a square section 200 mm long blast tube and the back clamp was mounted onto a pendulum via an adapter plate. According to Nurick et al. [27], the boundary conditions have a significant effect on the damage (such as large inelastic deformation) observed on panels subjected to uniformly loaded air blasts. Therefore, all panels were fully clamped to minimise the variability of the results due to boundary conditions, which is consistent with all prior work on these composites reported by Langdon and co-workers [10,11,28,29]. Explosive disks comprising 40 mm diameter PE4 plastic explosive were mounted on a on a polystyrene pad at the open end of the tube, ensuring a consistent stand-off distance. Air-blast loading was generated by detonating the disk-shaped PE4 explosive charge at the open end of the tube. The blast tube was employed to increase the spatial uniformity of the loading and to ensured that the impulse applied directly to the panels and that measured by the pendulum were the same. A similar test arrangement was used in Refs. [28,29]. The impulse was determined from the swing of the single-degree-of-freedom pendulum using a laser displacement sensor. The blast test arrangement is shown in Figure 7.

For selected tests where the risk of fragmentation was low, high-speed stereo-imaging equipment was mounted to the pendulum to film the panel response, following reference [30]. The back surface of the middle third of the panels was painted with a black and white speckle pattern. Two high-speed monochrome IDT NRS4 cameras (filming at 30 kfps with an exposure time of 31 µs) and LED lighting were positioned to provide a clear field of view of the central strip. The stereo-imaging system was calibrated to find the system projection and distortion parameters. A 19 × 19 pixels subset size and a grid spacing of 4 px were maintained for all tests. Digital image correlation (DIC) software was used to process the images to obtain the out of plane transient displacement across the mid-line of the panels.

### 5.2. Initial Observations

Six blast tests were performed on the GFRP laminate panels, and six tests were performed on the sandwich structures with the thicker face sheet faced the explosive. There was a general linear trend of increasing impulse with increasing charge mass, as expected. The front and back surfaces of the panels were inspected and photographed. The sandwich panels were sectioned along the mid-line to examine the internal damage to the core. Delamination area and crack lengths were measured. The results are summarised in Table 4. It appeared that the addition of the core improved the blast resistance of the sandwich structure at high impulses as less total cracking in the exposed areas and delamination were observed for a similar impulse GFRP blast tested panel as shown in Figure 8. The laminate G19/P20-6 appeared to have a low delaminated area and total crack length in the exposed area. Upon further investigation of the panel, extensive cracking was found in the clamped area and at the clamp boundary, and this is not recorded in Table 4. When cracks are initiated in other locations (such as the clamped region) it relieved stress in other parts of the structure leading to reductions in cracking in the exposed area as observed experimentally in this case. While we cannot be definitive in the reasons for why cracks should appear more extensive in the clamped region in this particular test, this is likely to be due to inherent variability of composite manufacture and the existence and distribution of small invisible defects in the clamped part of the structure. Furthermore, the measured permanent mid-point deflection was greater for the 35 g G19/P20-6 panel (1.2 mm) compared to the 40 g G19/P20-2 (0.6 mm) panel.

#### 5.2.1. Laminates

Photographs of the front and back surface of a laminate are shown in Figure 9. The laminates were slightly translucent, so damage or discolouration on one surface may be visible (but slightly obscured) on the other side. The front surfaces were discoloured by the residue from the blast products. The front surfaces of G19/P27-8 and G19/P27-9) were painted black (due to the translucent nature of the panel) for use with the DIC/stereo-imaging system.

The whitening of the panel along the edges of the exposed region indicated delamination. The extent of boundary delamination varied along the edge, being most extensive mid-boundary, matching the expected in-plane strain distribution along the edge. Delamination extended into the clamped boundary region, as shown in Figure 9. The panel area affected by delamination (not accounting for how many layers were delaminated, which could not be determined by visual inspection) increased with increasing impulse.

Matrix damage, delamination, and some fibre breakages (cracking) were evident at the edge of the clamped boundary, with selected close-in photographs of typical boundary failures shown in Figure 10. Cracking was also observed on the front exposed area at high impulses.

#### 5.2.2. Sandwich Panels

Photographs of the front and back surfaces of selected blast-tested sandwich panels are shown in Figure 11. Again, the glass fibre layers were slightly translucent, however overall, these panels were opaque. The front surfaces were discoloured by the residue from the blast products. The back surfaces of the sandwich panels were painted and speckled for use with the DIC/stereo-imaging system, which was subsequently removed after testing.

Similar to the laminates, whitening (indicating delamination) was found on the back face sheet of the panels. In general, the damage (such as delamination, boundary damage and cracking at the bolt holes) on the back face sheet resembled failure modes observed on the laminates. Localised delamination around the cracks were found on the front face sheet similar to that found in Ref. [6]. Little to no delamination was found extending from the clamped boundary on the front face sheet unlike the back face sheet and laminates. The panel area affected by delamination in Table 4 reflected that found on the back surface and summarily did not take account for which layers had delaminated. Delamination was found to increase with increasing impulse.

Cracking as a failure mode was initially observed on the front surface of G12FG6-1 at 39.8 Ns, where the cracks appeared to extend from the bolt holes to the exposed area. With increasing impulse, the cracking increased (both the number of cracks and length) and was also observed on the back surface of the panels at an impulse of 57.9 Ns and above. Greater crack lengths in the exposed area were observed on the front face sheet compared to the back face sheet. Photographs of cracking, both in the clamped area and exposed area, are shown in Figure 12.

Matrix failure was initially observed along the boundary on the back surface of G12FG6-1 (see back surface of G12FG6 in Figure 11). However, this damage was termed as boundary damage. With increasing impulse, matrix failure was also found in the exposed area of the sandwich panels as shown in Figure 12b.

As expected, there was a general trend where the thickness of the sandwich panel in the centre reduced with increasing impulse as shown in Figure 13. At the highest impulse tested at 84.5 Ns, the thickness of G12FG6-5 (35 g) was reduced by 8%. Figure 14 shows the comparison of the cross section along the midline of two sandwich panels, one of which did not appear to have significant damage after testing (G12FG6-6, 11 g). The core of G12FG6-5 (35 g) was crushed due to the blast load and at the edges, core cracking and slight fragmentation were observed. In all the tests performed, no debonding between the face sheets and core was observed. Additionally, it did not appear that either face sheets had significant permanent deformation due to the elastic nature of the composite. The lack of face sheet/core debonding will be of interests to modellers seeking to reduce the complexity of their models, as this appears to be a failure mode that could be neglected for similar cases.

## 6. Transient Response

### 6.1. Laminates

The transient mid-point displacement time histories of the laminates are shown in Figure 15. The transient response was elastic, with an initial steep rise in displacement followed by viscously damped oscillations (for at least 3 cycles). Furthermore, there was no significant permanent displacement. The natural period of vibration (based on the first two full oscillations) for the two panels were 3179 and 3684 rad/s for G19/P27-8 and G19/P27-9, respectively.

The transient displacement profiles for the GFRP panels tested are shown in Figure 16. A global dome-shaped profile was observed where the peak was found in the central region. The profiles were similar and appeared to overlap each other. Additionally, it seemed symmetric about the y-axis of the centre of the panel.

### 6.2. Sandwich Panels

The transient mid-point displacement time histories of the back faces of the sandwich panels are shown in Figure 17. Both types of materials exhibited high peak displacements. However, for the sandwich panel, while there was an initial steep rise, the subsequent fall in displacement was greater than the displacement observed in the GFRPs. During the initial rise, compression loading in the core would suppress the formation of core cracking despite the high shear loads generated. The rebound possibly caused the core to go through tension after the initial compression, which may result in core thickness recovery and, while not observed on the panels tested, interface bond failures. The minor damage on the cores observed from these tests also suggested that the rebound tensile stress did not exceed the tensile strength of the core.

Additionally, the following peak displacement observed appeared similar for the different sandwich panels and lower than the laminates despite different impulses. A comparison of the time histories between a laminate and sandwich panel tested at similar impulses is shown in Figure 18. The peak displacement is very similar in both panels, with a possible slight delay in the sandwich panel (due to the additional time taken for the waves to traverses the increased panel thickness and extra interfaces). The post-peak oscillation response of the back face of the sandwich structure seems more complex than the relatively simple viscously damped elastic vibration in the laminate tests, as load from the blast was not directly applied to the back face sheet. It was transferred to the back face sheet after the stress wave had passed the front face sheet, through the core and the core was compressed. Therefore, the intensity of the load transmitted would be dependent on the stiffness of the front face sheet and core compressive properties. The load would be dispersed differently between the front face sheet and back face sheet. It was postulated that different deflections for the front and back face sheet would be observed, unlike the almost identical front and back face sheet mid-point deflections found by Wang and Shukla [7] on symmetric sandwich panels. However, there posed some challenges in obtaining transient measurements on the front face sheet in blast testing to determine the relative displacement.

The transient displacement profiles for the sandwich panels tested are shown in Figure 19. A key difference in the displacement profiles between the two materials was profile 1 (indicated by the blue line), when the panel initially deforms. Unlike the GFRP, the mid-section of profile 1 was flat. Once the panel reached its peak displacement, the profiles were global dome-shaped and resembled profiles found in the GFRPs.

## 7. Sequence of Failure and Failure Mechanisms

At the impulses tested, complete failure did not occur in the panel due to the blast. Furthermore, the pressure distribution from the blast load was considered uniform. Therefore, the response of the sandwich panels can be described by three phases of motion [31]:a)Phase I: through-thickness wave propagationb)Phase II: transverse wave propagation, andc)Phase III: elastic vibration

No global deflection had yet occurred in Phase I, a compressive stress wave propagated from the incident (front) face sheet, through the core, to the rear face sheet. Core crushing would occur in this phase at stresses greater than the yield stress of the core. In comparison to the core, the face sheets were considered stiff and likely to remain elastic during the wave transmissions. Due to the restraint at the boundaries, impulsive transverse shear reaction forces would be induced. At the end of Phase I, the momentum and kinetic energy were transferred to the panel globally. Moreover, the impulsive shear reaction forces at the boundaries began to propagate towards the centre of the panel. Bending and shear loads developed behind the transverse stress wave front. Once the stress wave had reached the centre, the panel deflected to its maximum. At the end of Phase II, the plate started to rebound, leading to flexural oscillations in Phase III.

Assuming that most of the damage occurred in Phase I and Phase II, the probable sequence of damage for the asymmetric sandwich panels would be as shown in Table 5 and illustrated in Figure 20 and Figure 21.

## 8. Discussion

A failure initiation chart, shown in Figure 22 was used as indicators of the onset of a failure mode for a range of charge masses, similarly done in Ref. [13]. Dotted lines between different failure modes were used to show the combination of failure modes at higher charge masses.

In comparison to the laminates, back face sheet delamination initiated at a higher charge mass (of 25 g). Additionally, boundary damage and delamination were observed at similar loads on the GFRPs. Yet, there was a significant difference when the two types of damage were observed on the back face sheet. Cracking and boundary damage on the front face sheet occurred at the same charge mass as the Glass (19 layers)/Prime 27 laminates. Cracking also only occurred on the front surface of laminates. It might be possible that the cracking was caused by compressive loading as the panel deformed from a stress concentration spot (like a bolt hole or imperfection in the composite) and could be similar to face sheet wrinkling. For the sandwich panel, the critical stress for wrinkling was dependent on the core stiffness as well as the face sheet properties which was further described in Ref. [9].

Due to the load dispersing through the sandwich panel, it was expected that the initial damage would occur on the front face sheet and core. However, prominent boundary damage was first observed on the back face sheet (See Figure 11). Only slight damage (a single crack on one bolt hole) was observed on the front face sheet at the same impulse. The stress transmitted onto the back face sheet was large enough to cause significant damage than what would be expected on a symmetric sandwich panel, due to the thickness (and strength) of the back face sheet, along with the boundary opposing the deformation. As the load increased to 20 g, more damage, like cracking and core crushing, was observed on the front face sheet and core.

Less damage was evident in the sandwich panels compared to the laminates when subjected to the same charge mass detonation. Additionally, delamination occurred at a higher charge mass on sandwich panels compared to the laminates. While damage on the back face sheet was more significant (compared to symmetric sandwich panels as described in Refs. [6,7,8]), considering the core had a long low-level stress plateau in its constitutive behaviour, the applied load from the blast on the front face sheet was far greater than the load applied on the back face sheet. Therefore, it was postulated that it was more effective to increase the thickness of the front face sheet for better blast resistance. For these reasons, asymmetric sandwich panels had a better blast resistance compared to laminates and, very likely, symmetric sandwich panels of a similar mass when tested under uniform blast loading conditions.

## 9. Conclusions

The uniform blast loading response of sandwich panels consisting of E-glass fibre reinforced epoxy face sheets with two different number of plies (for the front and back face sheets) and 25 mm thick 80 PVC foam were investigated. A series of blast tests was conducted by detonating PE4 discs at a stand-off distance of 200 mm to the sandwich panel. The asymmetric nature of the panel influences the stress distribution as panel bends while providing additional thickness for blast protection, in comparison to that of a symmetric sandwich panel with the same total number of plies.

Due to the elastic nature of the composite, very little residual deflections on the laminates and glass fibre face sheets were observed. However, increased levels of damage (such as cracking, core compression and delamination) were evident in the sandwich panels as the impulse increased, and similarly for the laminates (such as delamination and cracking).

The results revealed a significant difference between the transient response of the laminates and the sandwich panels, particularly after the initial peak deflection. In the former case, the laminates behaved as viscously damped elastic vibrations which was not observed in the sandwich panels. The combined effect of the differences in stiffness and speed at which the stress wave travels between the three parts of the composite changes the deformation behaviour of the back face sheet. Quantitative and modelling investigations should be conducted to further understand the load transfer and transient behaviour within the sandwich constructure with varying face sheets.

## Figures and Tables

**Figure 1 materials-14-07118-f001:**
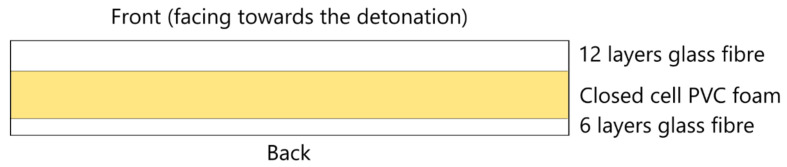
Cross section schematic of asymmetrical sandwich panel.

**Figure 2 materials-14-07118-f002:**
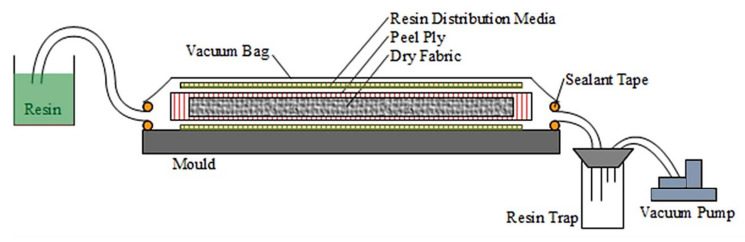
Schematic of VI process.

**Figure 3 materials-14-07118-f003:**
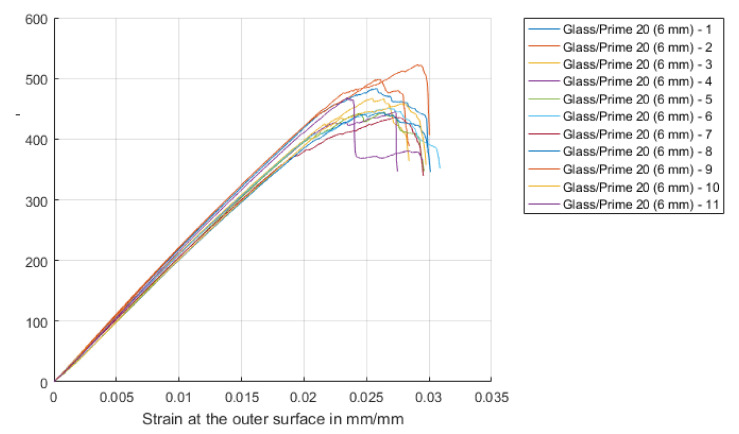
Stress-strain curves obtained from quasistatic three-point flexure tests on GFRP specimens.

**Figure 4 materials-14-07118-f004:**
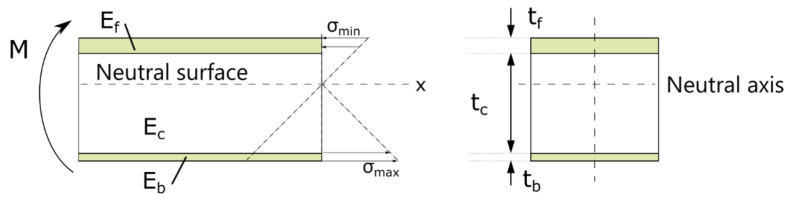
Stress distribution of sandwich beam under pure bending.

**Figure 5 materials-14-07118-f005:**
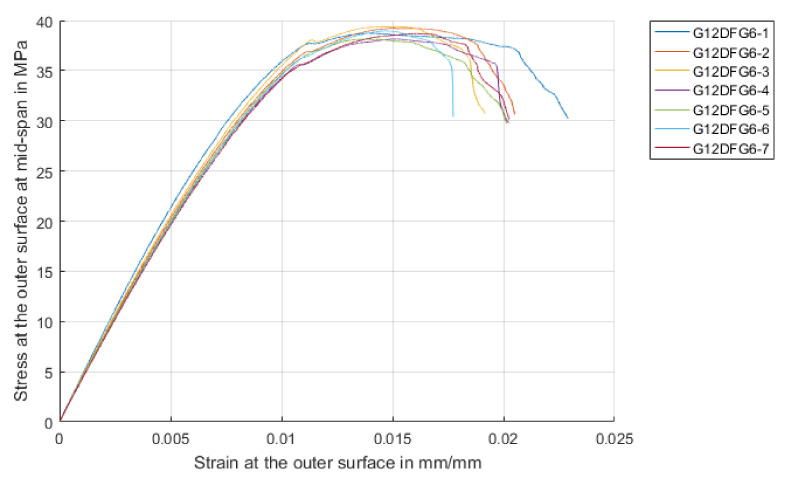
Stress-strain curves obtained from quasistatic three-point flexure tests on sandwich specimens.

**Figure 6 materials-14-07118-f006:**
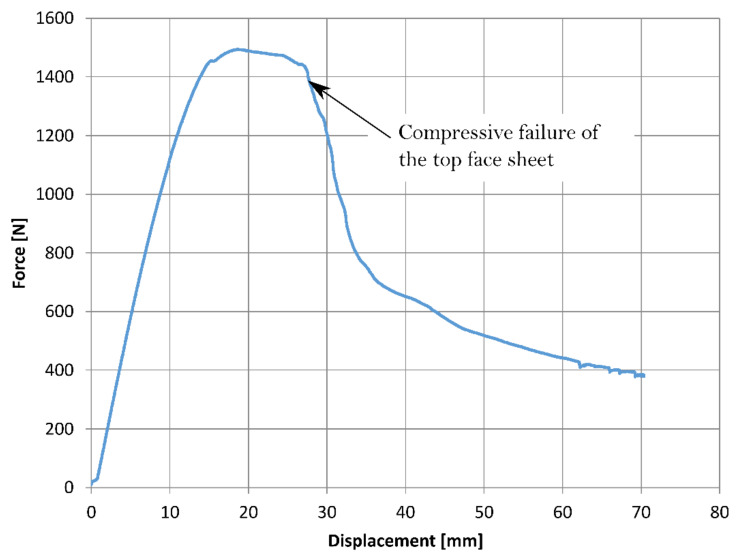
A typical force displacement curve obtained from three-point bend tests on sandwich beams.

**Figure 7 materials-14-07118-f007:**
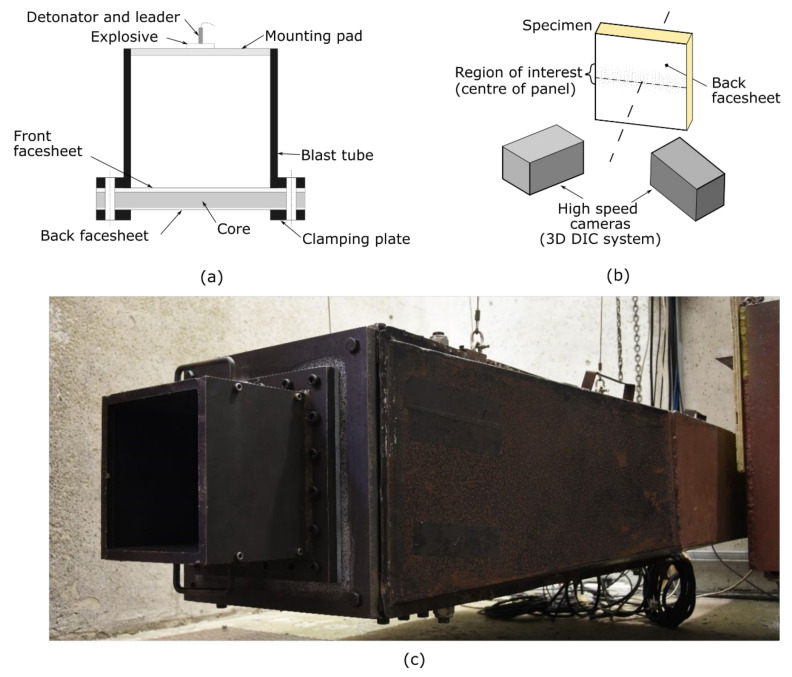
Schematic of blast experimental arrangement: (**a**) blast arrangement; (**b**) DIC testing arrangement; and (**c**) photograph of enclosed DIC-modified pendulum.

**Figure 8 materials-14-07118-f008:**
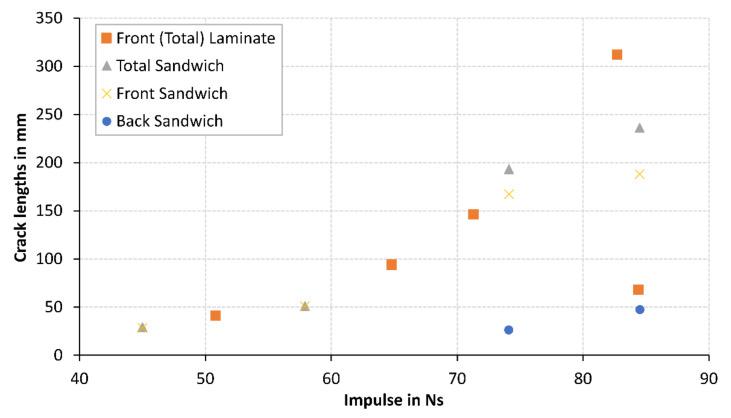
Graph of crack length within exposed area vs. impulse.

**Figure 9 materials-14-07118-f009:**
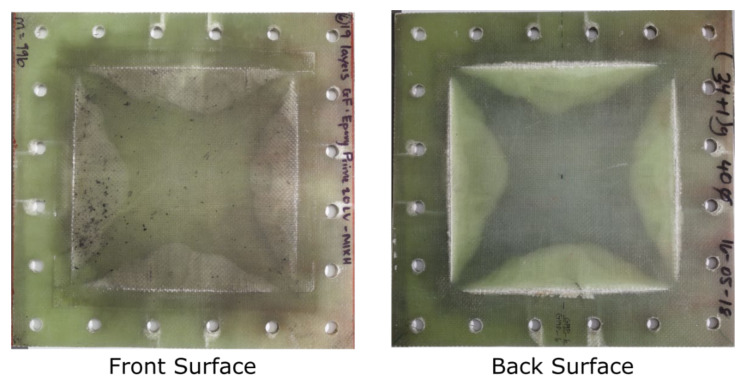
Photographs of front and back surfaces of G19/P20-6 laminate panel subjected to an impulse of 84.4 Ns (35 g).

**Figure 10 materials-14-07118-f010:**
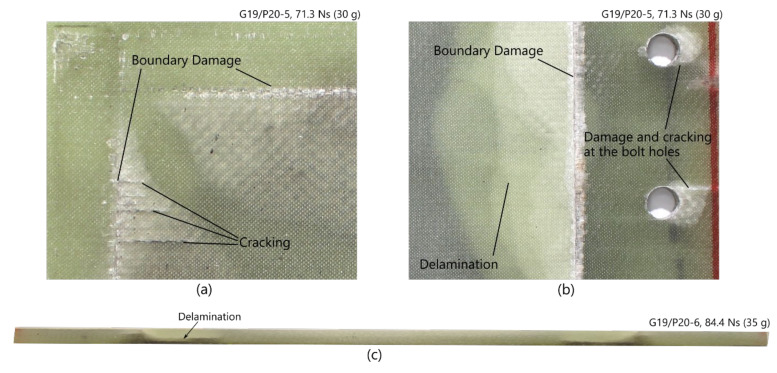
Damage (delamination, boundary damage, and cracking) observed in laminate panels on (**a**) front surface of G19/P20-5; (**b**) back surface of G19/P20-5; and (**c**) cross section along midline of G19/P20-6.

**Figure 11 materials-14-07118-f011:**
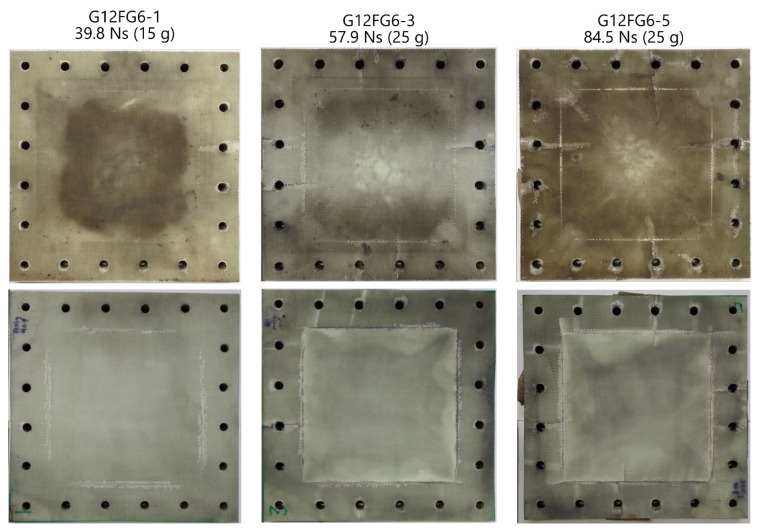
Photographs of front (**top row**) and back (**bottom row**) surfaces of selected sandwich panels.

**Figure 12 materials-14-07118-f012:**
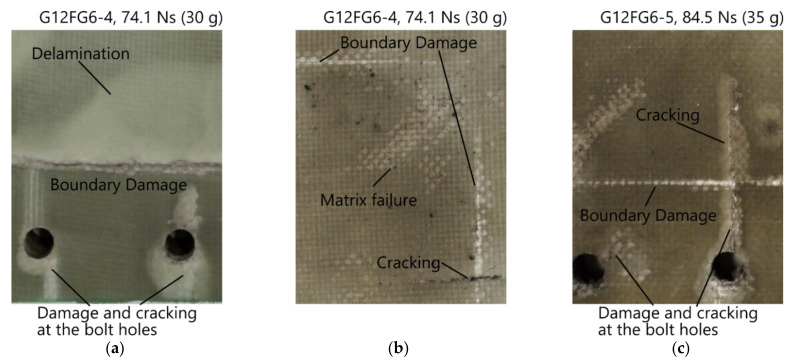
Damage observed in sandwich panels: (**a**) back surface of G12FG6-4 (30 g); (**b**) front exposed area of G12FG6-4 (30 g); (**c**) front surface of G12FG6-5 (35 g).

**Figure 13 materials-14-07118-f013:**
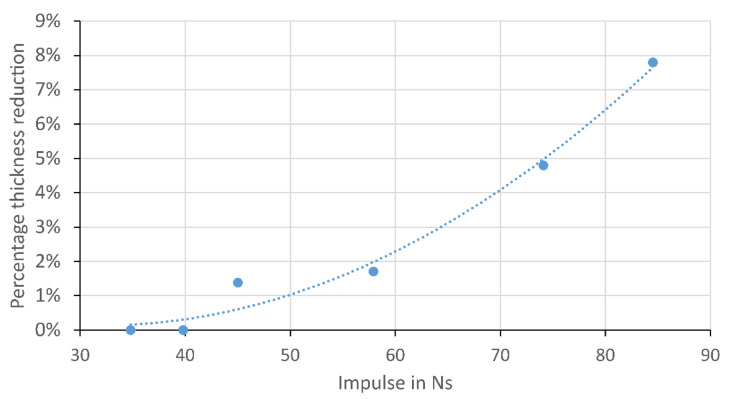
Graph of impulse versus percentage thickness reduction of blast-tested sandwich beams.

**Figure 14 materials-14-07118-f014:**
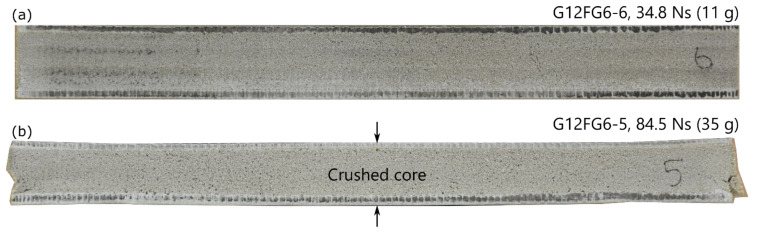
Cross section along midline of sandwich panels (**a**) G12FG6-6 (11 g) and (**b**) G12FG6-5 (35 g).

**Figure 15 materials-14-07118-f015:**
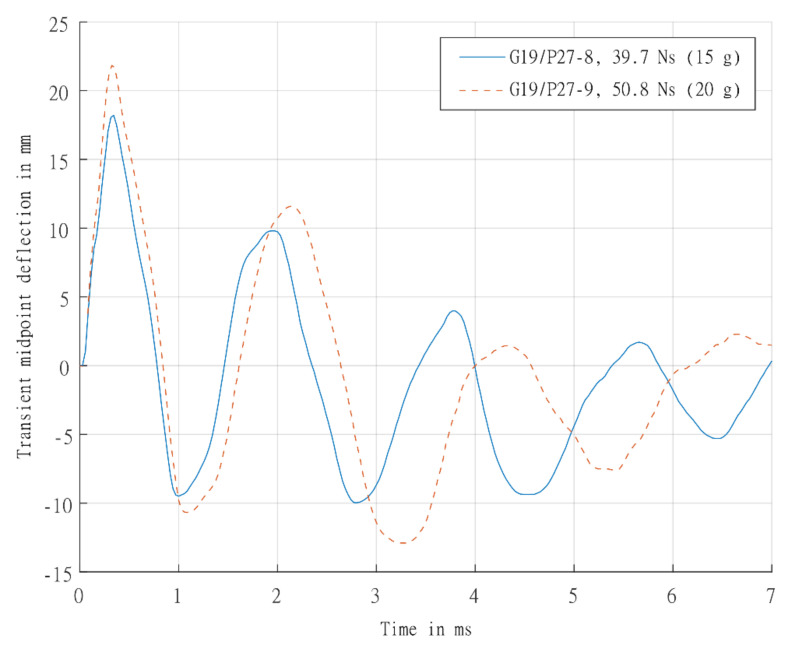
Graph showing experimentally determined transient mid-point displacement-time histories for laminate panels G19/P27-8 (15 g) and G19/P27-9 (20 g).

**Figure 16 materials-14-07118-f016:**
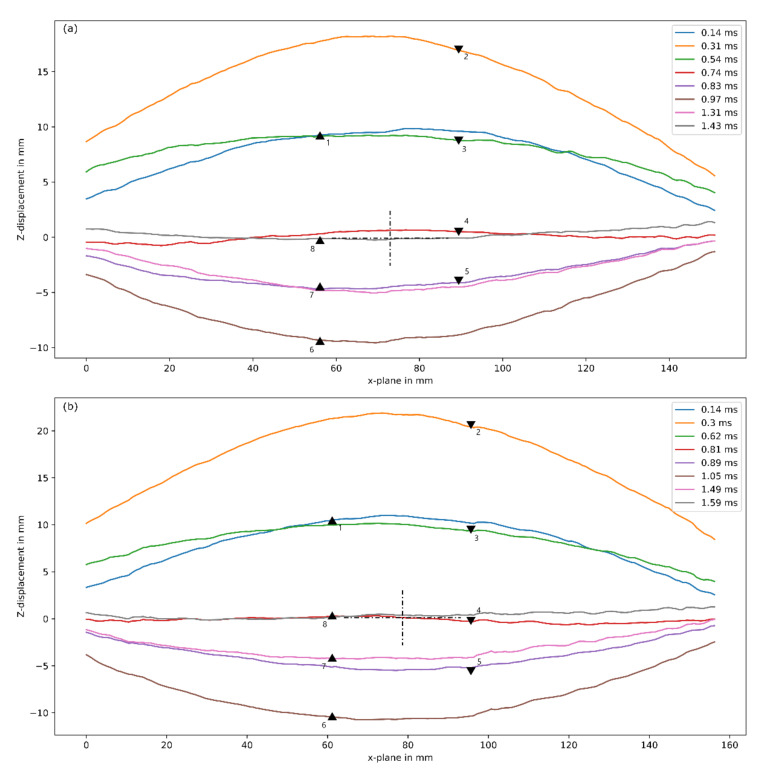
Transient deformed profiles of selected laminates (**a**) G19/P27-8 (15 g) and (**b**) G19/P27-9 (20 g) obtained from DIC analysis, where triangles indicate direction of motion.

**Figure 17 materials-14-07118-f017:**
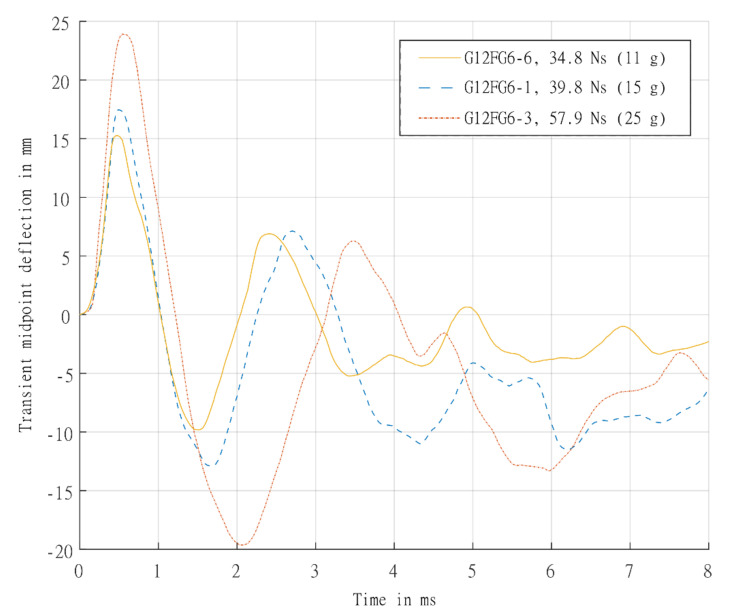
Graph showing back face sheet experimentally determined transient mid-point displacement-time histories for sandwich panels G12FG6-6 (11 g), G12FG6-1 (15 g), and G12FG6-3 (25 g).

**Figure 18 materials-14-07118-f018:**
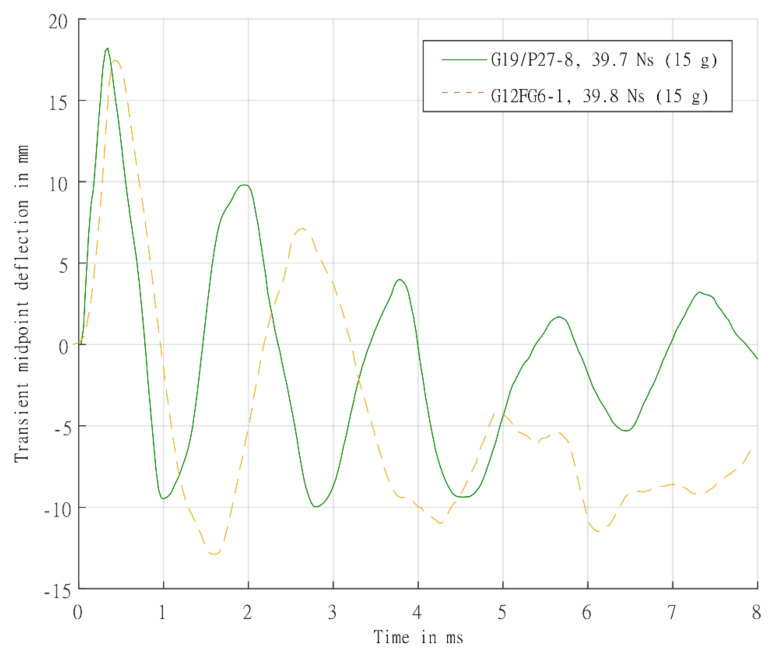
Comparison of experimentally determined transient mid-point displacement-time histories for back surface of laminate panel G19/P27-8 and sandwich panel G12FG6-1, both tested at 15 g.

**Figure 19 materials-14-07118-f019:**
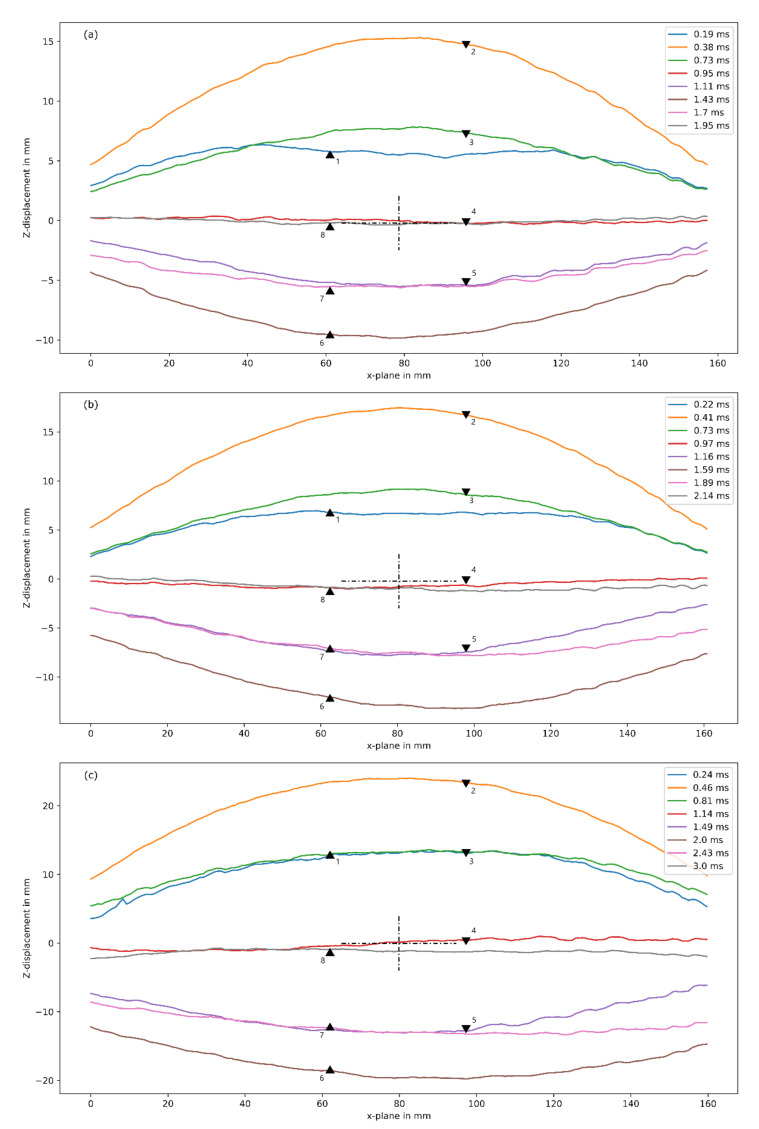
Transient deformed profiles of (**a**) G12FG6-6 (10 g), (**b**) G12FG6-1 (15 g), and (**c**) G12FG6-3 (25 g), obtained from DIC analysis, where triangles indicate direction of motion.

**Figure 20 materials-14-07118-f020:**
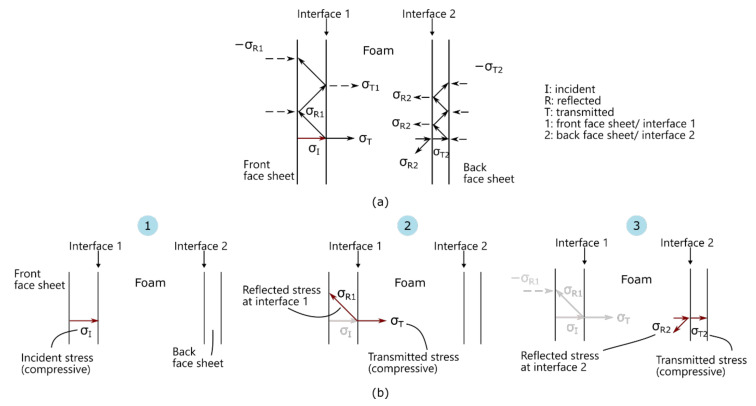
Schematic of Phase I: through-thickness wave propagation where (**a**) overview of transmitted and reflected waves in sandwich structure and (**b**) propagation of stress waves through different layers.

**Figure 21 materials-14-07118-f021:**
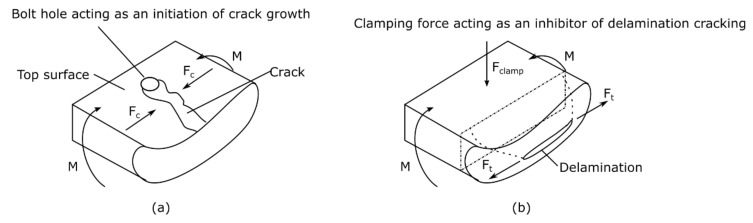
Reaction at boundaries of (**a**) front face sheet and (**b**) back face sheet due to global bending.

**Figure 22 materials-14-07118-f022:**
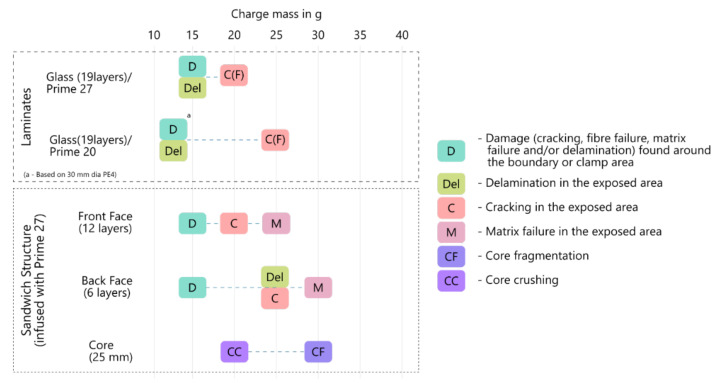
Failure mode initiation chart for laminates and sandwich structure tested.

**Table 1 materials-14-07118-t001:** Technical data for Divinycell H80 [22].

Property	Method	Value
Compressive strength	ASTM D 1621	1.4 MPa
Tensile strength	ASTM D 1623	2.5 MPa
Tensile modulus	ASTM D 1623	95 MPa

**Table 2 materials-14-07118-t002:** Summary of flexural test results.

	Flexural Strength, σ^f^_max_ [MPa]		Effective Flexural Modulus, E^f^ [GPa]		Strain at Failure, ε^f^_fail_ (%)	
	Mean	Std Dev	Mean	Std Dev	Mean	Std Dev
GFRP laminates	452	12	21.3	0.6	2.9	0.1
Sandwich structure	39	0.8	3.9	0.1	2.1	0.2

**Table 3 materials-14-07118-t003:** Summary of in-plane tensile test results.

	Modulus of Elasticity, E^T^ [GPa]		Apparent Elastic Limit, σ^T^_Y_ [GPa]		Ultimate Tensile Strength, σ^T^_U_ [MPa]		Strain at Failure, ε^T^_fail_ (%)	
	Mean	Std Dev	Mean	Std Dev	Mean	Std Dev	Mean	Std Dev
0/90°	20.4	0.3	164.0	27.7	359	9.0	1.96	0.06
45°	6.7	0.7	45.5	2.2	110	2.0	>80	-

**Table 4 materials-14-07118-t004:** Summary of blast test results.

Material	Test #	Thickness (mm)	Charge Mass (g)	Impulse (Ns)	% Delaminated Area ^a^	Total Crack Lengths ^b^ (mm)	DIC Avail.
19-layer GFRP laminate panels	G19/P27-8	6.0	15	39.7	52	0	X
	G19/P27-9	6.0	20	50.8	72	41	X
	G19/P20-3	6.2	25	64.8	51	94	
	G19/P20-5	6.2	30	71.3	54	146	
	G19/P20-6	6.1	35	84.4	47	68	
	G19/P20-2	6.2	40	82.7	59	312	
Sandwich panels	G12FG6-6	30.8	11	34.8	0	0	X
	G12FG6-1	30.9	15	39.8	0	0	X
	G12FG6-2	30.9	20	45	21	29	
	G12FG6-3	30.9	25	57.9	40	51	X
	G12FG6-4	30.7	30	74.1	54	193	
	G12FG6-5	30.8	35	84.5	55	236	

^a^—Delamination found and based on 200 mm × 200 mm exposed area; ^b^—cracking found within 200 mm × 200 mm exposed areas of back and front surfaces.

**Table 5 materials-14-07118-t005:** Proposed sequence of damage in an asymmetric sandwich panel.

Phase I	Through-thickness incident stress wave transmitted in the front face sheet as shown in Figure 20b(1) [31,32]. Initial compression of the (foam) core due to the transmitted compressive stress wave from the blast shown in Figure 20b(2) [31]. Boundary damage occurred on the back face of the sandwich panel due to the compressive stress wave and impulsive transverse shear reaction forces (Figure 20b(3)). Delamination also possibly occurred due to reflected tensile stress waves in the face sheet shown in Figure 20a.
Phase II	4. The front face deformed, subsequently caused the core and rear face to deform. 5. Bending and shear loads, as the panel deformed, caused damage in the different parts of the sandwich panel. Primarily, delamination extending from the boundary towards the centre occurred on the rear face sheet (Figure 21a) due to its thickness (relative low strength and stiffness). At high impulses, compressive stresses on the front face sheet appeared to cause cracking and matrix failure, extending from high stress concentration areas such as bolt holes and clamped boundary as shown in Figure 21b. Further core compression would occur due to high shear loads.
Phase III	6. The differences in stiffness of the face sheets influenced the reduction of deflection and energy dissipated. As shown in DIC, response would deviate from elastic vibration, but oscillations would still occur. 7. As the panel rebounded, boundary damage occurred on the front face sheet due to shear reaction forces. Compressive stresses on the back face sheet caused cracking from the bolt holes. Core went through tension which might result in core thickness recovery.

## Data Availability

The data presented in this study are available on request from the corresponding author.

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
