# Peer review of "Towards an Understanding of the Effect of Adding a Foam Core on the Blast Performance of Glass Fibre Reinforced Epoxy Laminate Panels"

_materials, 2021, doi:10.3390/ma14237118_

Round 1

Reviewer 1 Report

This paper is sound and well presented

Author Response

We thank the reviewer for their positive comments. 

Reviewer 2 Report

This is an interesting work about the blast performances of sandwich structures with different skin thicknesses. The paper is well written, methods are properly described and experimental results adequately discussed. I have just one concern about the results from static flexural tests:

It is not clear how was the stress for the sandwich panel estimated, and probably is should be better explained. In any case, it is very strange that the modulus for the sandwich panel is so low compared to the modulus for the laminate. The authors should give more explanation on this section

Author Response

Thank you for the positive comments on the paper. We calculated the smeared flexural properties of the sandwich panel to compare with the laminates. To prevent confusion, we have renamed the smeared modulus obtained from the static flexural tests as the effective flexural modulus in Table 2. This modulus is lower for the sandwich panel, as the second moment of area of the sandwich panel is higher. Additionally, there is less glass fibre/epoxy layers in the sandwich panel.

Reviewer 3 Report

A series of experiments were performed to prove the advantage of the asymmetric arrangement of sandwich panels against blast. However, the results were compared to laminates, not to sandwich panels with symmetric layer arrangement of laminates.

Also, it was claimed that the mass difference was negligible between G19 and G12/foam/G6. The difference was 0.2 kg, which means that the sandwich structure should have been compared to laminates consisting of 22 layers (instead of 19), or the thickness of the core (foam) should have been reduced to 5-6 mm. Since the mass affects the mechanical properties, the effect of the extra 0.2 kg (20% difference) is considerable.

It seems that the way of clamping in the blast test influences the detected failure modes. (e.g., the appearance of cracks) Thus the failure mode initiation chart (Fig. 25) is based not only on the properties of the investigated materials.

Others:

Sometimes the charge mass is used as the determining parameter (i.e., Fig. 25 ), sometimes the impulse (i.e., Fig. 11).

Does the total crack length not depend on the way the panel is fixed?

According to Table 4, if the charge mass is 35 g, it is better to use G19 than the sandwich panel. Why?

In the case of G19, the charge mass changed from 15g to 40 g, while for G12/foam/G6, it was between 11g and 35g. Why were different circumstances used?

Flexural, tensile, and compressive properties were measured but not used in the explanation.

Many statements in Table 5 are not proved. E.g., if there is “initial compression of the foam” in the first phase, why is the transient mid-point displacement maximum the same for G19 and G12/foam/G6?

Reviewer 4 Report

This research discusses the transient behavior of the foam-core multi-layered sandwich structures, when subjected to air-blast loading.

The topic is not novel, there have been conducted previous works, including by the authors of this paper.

The topic may be considered original when referring to the subject of the study: asymmetric sandwich panels with polymer foams and FRP face sheets.

The experimental results may be useful for researchers interested on this subject; they could be used for benchmarking in the attempt of developing analytical or numerical models.

In general, the conclusions consistent with the evidence and arguments presented. However, they may be improved by including, at each section, further explanations regarding the results obtained.

Overall the main question posed been addressed; this study opens new areas of investigation of the subject by developing analytical/ numerical models.

The paper investigates the blast response of a sandwich panel consisting of glass fiber epoxy face sheets with different thickness and foam as a core. The research is done using the experimental approach, experiments being conducted in an adequate way and results clearly presented. However, the experimental results obtained and presented in the paper could have been better valued by complementing the research with the development of an analytical/ numerical model.

Author Response

Thank you for your comments. Ideally, it would be nice to incorporate either some analytical or numerical simulations to complement the experimental data. However, in its current form, the paper is quite lengthy. Moreover, we had not planned to include any analytical/simulation work given the time frame that we had to submit this manuscript. We, however, feel that we have presented all the required data for the readers to perform any analytical or numerical simulations as future work should they wish to.

Round 2

Reviewer 3 Report

There have been made only minor changes despite the suggestions.

1) It seems that the way of clamping in the blast test influences the detected failure modes. The authors agreed with this statement, however, the abstract was not changed accordingly.

2) Flexural, tensile, and compressive properties were measured but not used in the explanation.
"The material properties presented was to confirm consistency of manufacture, and to provide properties for future modelling and simulation work."

If these results were presented only to confirm consistency then a table containing the values and the standard deviation is enough instead of 4 pages of data without any explanation or only containing trivial remarks. E.g. Divinycell H200 is not used in the whole work, why did you put a stress-strain curve of that material in the paper? Similarly, Fig 2b is not related to this work, not used, why did the authors put it in the paper?

If the authors want to use the measured data for future work, please put the data in that paper, these are not related to this work.

(l. 216 modeling and simulation)

l. 244 the linear (or more precisely quasi-linear stage ends at ~25 MPa, please check it, however, as I've mentioned previously I strongly suggest omitting/rewriting 2.3,4.1,4.2 parts.

3) "According to Table 4, if the charge mass is 35 g, it is better to use G19 than the sandwich panel. Why?" The authors answered that they added l. 362-366 (please check the size of the letters), however, they just describe the phenomenon and they did not explain it. It is also possible that the sample had some invisible defects.

4) The statement l.436-441 can be the consequence of the "boundary condition".

others:

l.123 in Figure 1

l. 190 was completed

l. 374 Figure 12 is in bold

l.378 use of

l. 413 delamination area
